# The return to water in ancestral *Xenopus* was accompanied by a novel mechanism for producing and shaping vocal signals

Ursula Kwong-Brown[1], Martha L Tobias[1], Damian O Elias[2], Ian C Hall[1†], Coen PH Elemans[3]*, Darcy B Kelley[1]*

[1]Department of Biological Sciences, Columbia University, New York, United States; [2]Department of Environmental Science, Policy and Management, University of California, Berkeley, Berkeley, United States; [3]Department of Biology, University of Southern Denmark, Campusvej, Denmark

*For correspondence:
coen@biology.sdu.dk (CPHE);
dbk3@columbia.edu (DBK)

Present address: †Department of Biological Sciences, Benedictine University, Lisle, United States

Competing interests: The authors declare that no competing interests exist.

**Abstract** Listeners locate potential mates using species-specific vocal signals. As tetrapods transitioned from water to land, lungs replaced gills, allowing expiration to drive sound production. Some frogs then returned to water. Here we explore how air-driven sound production changed upon re-entry to preserve essential acoustic information on species identity in the secondarily aquatic frog genus *Xenopus*. We filmed movements of cartilage and muscles during evoked sound production in isolated larynges. Results refute the current theory for *Xenopus* vocalization, cavitation, and favor instead sound production by mechanical excitation of laryngeal resonance modes following rapid separation of laryngeal arytenoid discs. Resulting frequency resonance modes (dyads) are intrinsic to the larynx rather than due to neuromuscular control. Dyads are a distinctive acoustic signature. While their component frequencies overlap across species, their ratio is shared within each *Xenopus* clade providing information on species identity that could facilitate both conspecific localization and ancient species divergence.
**Editorial note:** This article has been through an editorial process in which the authors decide how to respond to the issues raised during peer review. The Reviewing Editor's assessment is that all the issues have been addressed (see decision letter).
DOI: https://doi.org/10.7554/eLife.39946.001

## Introduction

In the transition from water to land in early tetrapods, lungs replaced gills for respiration (*Daeschler et al., 2006*). Many current tetrapods use air movement to empower specialized vocal organs such as the larynx of frogs and mammals and the syrinx of birds (*Elemans et al., 2015*). The resulting sounds are shaped by a combination of vibrating elements and cavity resonances to voice different acoustic qualities that convey sex, age, species, emotional state and even intent (*Hall et al., 2013*). While voice provides essential information for social interactions, we know surprisingly little about how vertebrate vocal organs create these complex acoustic features. In particular, when an ancestral tetrapod leaves the land and vocalizes underwater without air movement, how are communication sounds produced and then shaped to maintain essential social information, and how do they diversify during speciation? Frogs in the secondarily aquatic genus *Xenopus* (*Evans et al., 2015*) present an informative system for addressing these questions.

In *Xenopus*, social communication is dominated by vocal signaling (*Kelley et al., 2017*). Males in each species produce distinctive advertisement calls underwater whose acoustic features inform species identity (*Evans et al., 2015*). These calls consist of a series of sound pulses that form species-typical temporal patterns and characteristically include two dominant frequencies (DFs) (*Hall et al.,*

**eLife digest** The voice is a unique characteristic that we use to identify one another – including someone's sex, age and mood. We speak by using air flow to vibrate our vocal folds, commonly known as vocal cords. The land-living ancestors of the African clawed frog *Xenopus* also used breath and vocal cords to communicate, but they returned to aquatic life 180 million years ago and had to evolve a different way to create sounds. Today's *Xenopus* live in water and use a new mechanism that lets them sing for hours underwater without coming up to breathe. Males from each major group of *Xenopus* species produce courtship songs with harmonic intervals corresponding to an octave, a perfect fourth, or a major or minor third.

Today's *Xenopus* species do not have any vocal cords. Instead, they have an elaborate set of vocal components: the muscles of the larynx contract paired, movable rods that end in discs. For the past 40 years, it was thought that these frogs create sounds by collapsing small air bubbles between the discs, similar to snapping shrimp. But such bubbles have never been observed, and exactly how these frogs manage to create sounds underwater has been a mystery.

Here, Kwong-Brown et al. filmed the larynx as it was stimulated to produce sounds and discovered that the rapid separation of the discs excites the larynx and the surrounding tissues to create the harmonic frequencies. Then, to determine how the frog creates its harmonic intervals, Kwong-Brown et al. tried to 'detune' the larynx. In a series of experiments, they placed weights on the surface of the larynx, drilled a hole in the cartilage and filled it with helium, or introduced small glass beads. None of these attempts had any effect. However, rupturing the elastic cartilages within the larynx – which separate its internal cavity into three chambers – disrupted the harmonic intervals.

This new way of creating underwater sounds helped to maintain the quality of the frog's voice and may explain how *Xenopus* can shape its songs to convey crucial information to others, such as identifying species, sex and social intent.

DOI: https://doi.org/10.7554/eLife.39946.002

*2013*; *Tobias et al., 2011*). The sound pulses that comprise *Xenopus* calls are produced in the larynx, (*Tobias and Kelley, 1987*) a vocal organ interposed between the nasal and buccal cavities and the lungs (*Figure 1A*). Vocal folds are absent (*Ridewood, 1897*) and a separate glottis gates air flow to and from the lungs (*Brett and Shelton, 1979*). The larynx consists of a cricoid frame or 'box' of hyaline cartilage flanked bilaterally by bipennate muscles. These insert anteriorly, via a tendon, onto paired, closely apposed arytenoid cartilage discs (*Figure 1B*; *Figure 1—figure supplement 1*) whose medial faces are coated by mucopolysaccharide secreted by adjacent cells (*Yager, 1992*). The discs are suspended in elastic tissue composed of elastic cartilage and elastin fibrils (*Figure 1— figure supplement 1*) (*Yager, 1992*). Electrical stimulation of laryngeal muscles or nerves results in species-specific sound pulses, both in situ and ex vivo (*Yager, 1992*; *Tobias and Kelley, 1987*). Sound is thus produced without air flow or vocal folds. In *X. borealis,* separations of the paired arytenoid discs accompany sound pulses, (*Yager, 1992*) but how disc motion results in sound production has not been resolved.

An unusual mechanism – implosion of air bubbles or cavitation (*Yager, 1992*) - is the currently accepted (*Irisarri et al., 2011*; *Ladich and Winkler, 2017*) hypothesis for underwater laryngeal sound production in *Xenopus.* In this scenario, the high velocity separation of the arytenoid discs causes formation of bubbles that then implode and produce sounds. Cavitation bubbles are known to produce hydrodynamic propeller noise (*Carlton, 2012*) and the 'snaps' of some species of shrimp (*Versluis et al., 2000*). However, *a priori* cavitation - creating "a bubble between the discs at a pressure below ambient . . . that . . . implodes as air rushes into the cleft at high speeds, producing a click" (*Yager, 1992*) - seems an unlikely cause of sound production in *Xenopus*. The small film of fluid between the arytenoid discs should allow neither high velocity flow nor bubble formation, and bubbles have not yet been observed. Additionally, cavitation bubble collapse produces a high amplitude pressure pulse (~50–100 kPa), several orders of magnitude louder than the radiated sound pressure of *Xenopus* sound pulses. Finally, the duration of pressure transients produced by the collapse of cavitation bubbles are in the microsecond range, rather than the millisecond range of *Xenopus* sound pulses.

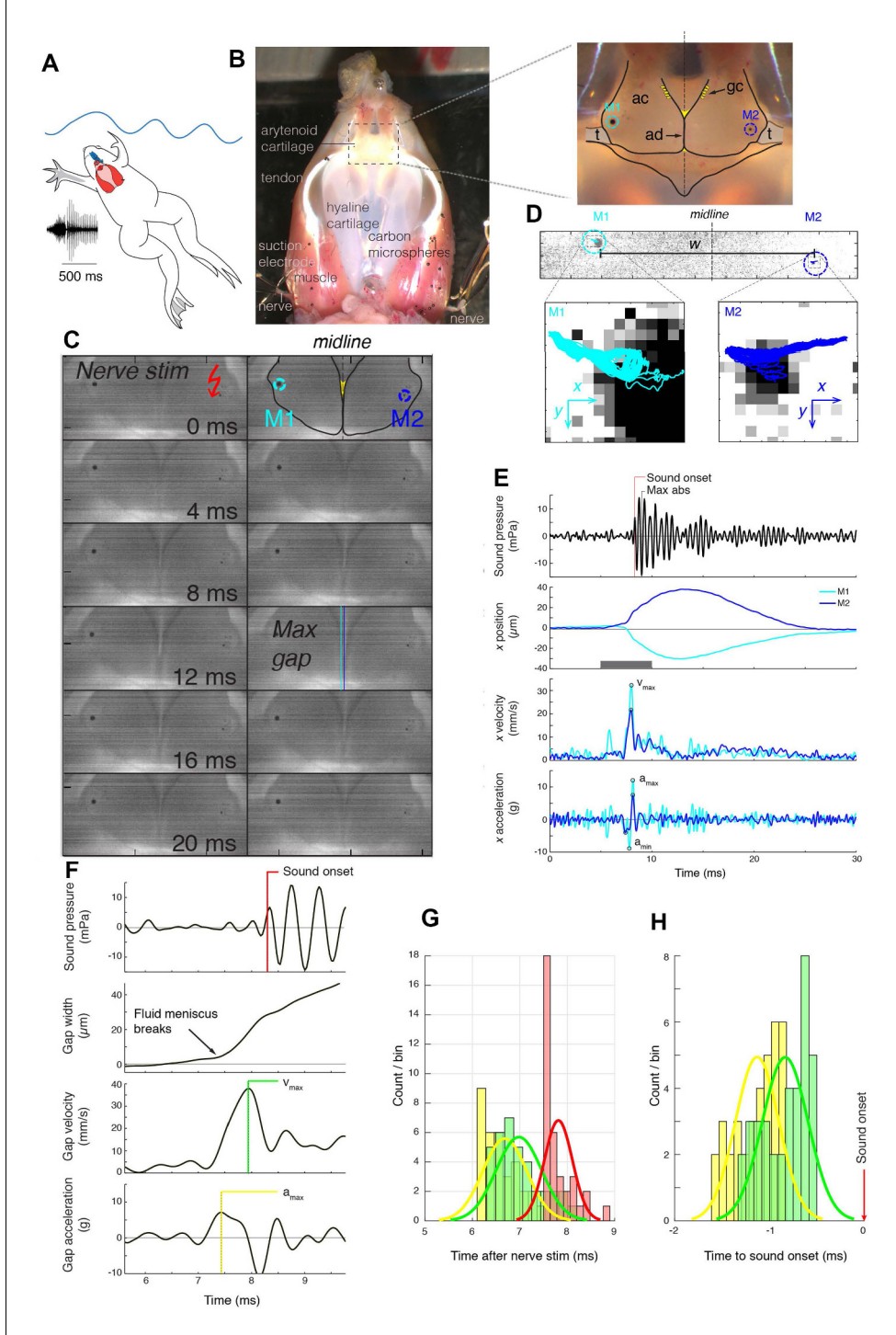

**Figure 1.** Arytenoid disc kinematics associated with underwater sound production in the ex vivo larynx of *Xenopus laevis*. (**A**) *Xenopus* call while submerged. A ventral view of a reproductively active, *X. laevis* male (nuptial pads in grey on the inner surface of the forearms), underwater (blue waves); larynx in red and more dorsal brain in blue. This view of the larynx is schematic (i.e. the dorsal rather than the ventral side is illustrated) in order to correspond to the actual isolated larynx in (**B**). On the left, an oscillogram (sound intensity vs time) of a single, biphasic call that includes a fast and slow trill. Each vertical line indicates a sound pulse; ~60 pulses/s for fast trill and ~30 pulses/s for slow trill. (**B**) Dorsal aspect of an isolated *X. laevis* larynx, a cricoid box of hyaline cartilage flanked by muscles. Each effective contraction/relaxation of these paired laryngeal muscles produces a single sound pulse. In the preparation illustrated, sound pulses are evoked by electrical stimulation of both laryngeal nerves via suction

*Figure 1 continued on next page*

*Figure 1 continued*

electrodes. Inset: Each muscle contraction produces a transient increase in tension on the arytenoid discs (ad) located within the arytenoid cartilages (ac) via the tendons (t). Globule cells (gc) secrete a mucopolysaccharide onto the medial surfaces of the arytenoid discs (*Yager, 1992*). Carbon microspheres (e.g. M1 and M2) placed on the surface of the larynx track muscle and cartilage positions. (C) Still photo of arytenoid cartilage motion filmed at 10,000 fps and illustrated at 2 ms intervals. Nerve stimulation occurs in top left image. (D) Upper panel: Higher magnification images of each bead in B) during filming at 44,000 fps. Lower panel: The small motion of each bead (cyan, left; blue, right) during 40 consecutive stimulations at 40 Hz. (E) Sound (top panel, corrected for time of flight) and the position, velocity and acceleration of the two beads (color coded as in D) during a single pulse. Nerve stimulation occurs at t = 0. (F) Kinematic data for gap width (w) in C) in relation to sound onset. While the precise onset (red line) is hard to determine due to acoustic noise, sound production follows peak velocity (green) or acceleration (yellow). (G) The timing of sound onset (red), gap peak acceleration (yellow) and peak velocity (green) during 40 consecutive clicks for one larynx relative to nerve stimulation. (H) Sound onset relative to gap peak acceleration (yellow) and peak velocity (green) during 40 consecutive clicks for the larynx in G).

DOI: https://doi.org/10.7554/eLife.39946.003

The following figure supplement is available for figure 1:

**Figure supplement 1.** The arytenoid discs are suspended in elastic cartilage.

DOI: https://doi.org/10.7554/eLife.39946.004

## Results

## Mechanism of sound pulse production

To empirically test the cavitation hypothesis, we filmed isolated *X. laevis* larynges during sound pulse production evoked by stimulation of the laryngeal nerve (*Tobias and Kelley, 1987*). As reported previously, (*Yager, 1992*) disc movements accompanied sound production (*Figure 1C,E,F*; *Video 1*).

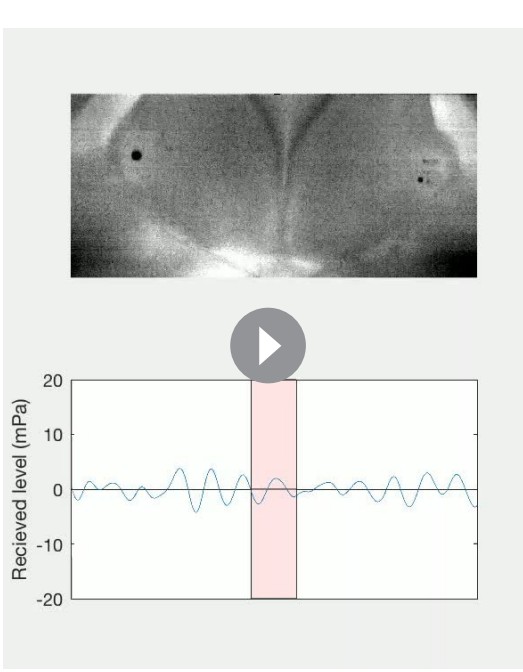

**Video 1.** Motion of arytenoid discs filmed at 10,000 fps during 40 Hz nerve stimulation. Video playback is slowed down 40x and resampled. The sound pressure waveform is displayed below the image and the sound associated with each frame is in the red area. Two carbon spheres can be seen on each arytenoid (large sphere about 80 µm diameter).

DOI: https://doi.org/10.7554/eLife.39946.005

To track the position of the arytenoid discs, we placed 40–80 µm carbon microspheres over the discs (*Figure 1B,C*) and computed disc position at subpixel resolution by interpolating the 2D intensity correlations of the spheres with each image (See Materials and methods, *Figure 1D*). This approach revealed disc position at speeds of up to 44,000 fps and allowed us to calculate disc velocity and acceleration profile in relation to sound pressure (*Figure 1E–H*).

Nerve stimulation first produces isometric contraction of the bipennate muscles during which the arytenoid discs remain apposed. In favorable preparations, a fluid layer could then be observed retreating from the medial surface of the discs with increasing isometric force. This observation suggests that the discs are kept together by a capillary binding force through a liquid bridge. When bilaterally exerted muscle force overcomes the binding force, the liquid bridge ruptures (*Figure 1F*) and the discs separate rapidly with mean gap peak acceleration of $13.5 \pm 9.1$ *g* (N = 3; range: 6.4–23.8 *g*, where 1 $g = 9.82$ m/s$^2$) and mean peak velocity of $50.3 \pm 21.9$ mm/s (N = 3; range: 35.5–75.4 mm/s) (see examples: *Figure 1E*; *Figure 2B*). Disc peak deceleration (10.6 $\pm$ 5.2 *g*; N = 3; range: 6.5–16.4 *g*) occurs only $0.57 \pm 0.10$ ms after peak acceleration. In this short time, the gap between the discs enlarges to $18.6 \pm 5.5$ µm

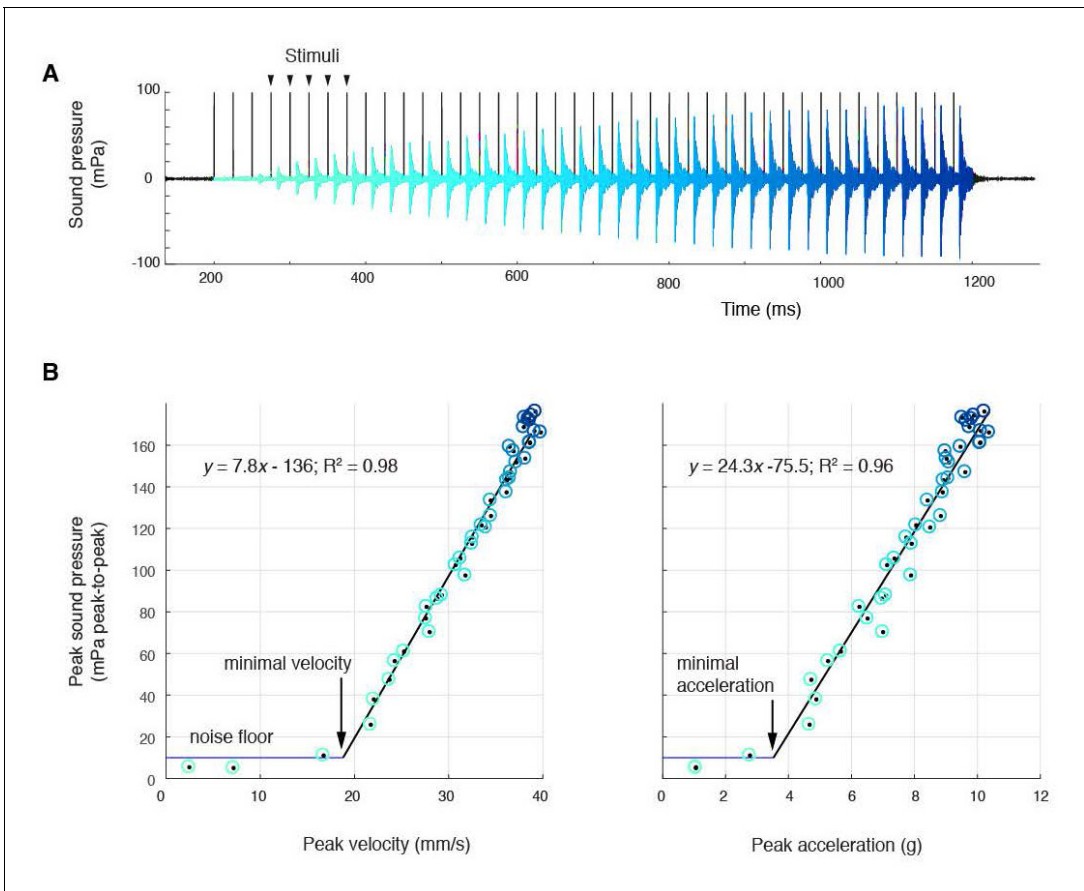

**Figure 2.** Sound amplitude correlates linearly with disc kinematics, (**A**) Sound pressure across 40 consecutive stimulations (black vertical lines) at 40 Hz (aqua, first stimulus; blue, last stimulus). (**B**) Peak sound pressure per stimulus (color-coded as in A) increases linearly with peak velocity (linear regression for three individuals: y = 5.2x-210.5, $R^2$ = 0.96; y = 1.8x-35.1, $R^2$ = 0.89; y = 7.8–136.3, $R^2$ = 0.98) and peak acceleration (linear regression for three individuals: y = 14.3x-106.9, $R^2$ = 0.91; y = 5.6x-6.1, $R^2$ = 0.89; y = 24.3–75.5, $R^2$ = 0.96). A minimal velocity and acceleration is required before sound is radiated

DOI: https://doi.org/10.7554/eLife.39946.006

The following figure supplement is available for figure 2:

**Figure supplement 1.** Injection of water into the supra-disc space via the glottis abolishes sound pulse production.

DOI: https://doi.org/10.7554/eLife.39946.007

(range: 14.1–24.8 µm), a value that is 27.5 ± 8.0% (range: 22.8–35.6%) of the maximum gap width of 72.6 ± 34.5 µm (N = 3; range: 39.6–108.4 µm). Peak acceleration and velocity precede sound onsets by 0.85 ± 0.33 and 0.51 ± 0.33 ms respectively (*Figure 1E–H*). The interval between disc separation and subsequent sound radiation reveals that these events are clearly associated. The delay (*Figure 1F*) between disc acceleration and velocity shows less variation (0.34 ± 0.06 ms, N = 3) indicating that sound onset timing after disc separation, rather than disc kinematics, is more variable between preparations.

At 40 Hz nerve stimulation rates (within natural call sound pulse rates (*Tobias et al., 2011*)), the first few stimuli do not result in sound pulse production (*Figure 2A*). This corroborates previous results that male laryngeal neuromuscular synapses are 'weak,' i.e., require facilitation to release sufficient neurotransmitter to generate a muscle action potential and contraction (*Tobias and Kelley, 1987*; *Ruel et al., 1997*). Subsequently, over multiple stimulations, peak-to-peak (ptp) sound pressure increases linearly with both peak velocity and acceleration and reaches a maximum received level of ~180 mPa ptp (at 44 mm) in all preparations (*Figure 2B*). Below a minimal peak velocity of 28 ± 12 mm/s and minimal peak acceleration of 4.6 ± 3.1 *g*, no sound is detected, suggesting a threshold disc velocity or acceleration required to produce sound. When water was introduced via

the glottis into the supradisc space, no sound was produced, corroborating earlier observations in *X. borealis* (*Yager, 1992*). Of disc gap width, peak velocity and peak acceleration, only disc peak velocity does not reach threshold when the liquid bridge holding the discs together is disrupted (*Figure 2—figure supplement 1*). This observation supports the hypothesis that a threshold disc peak velocity is required for sound production.

In summary, three sets of observations do not support the cavitation hypothesis: (1) we do not observe bubble formation, (2) sound pressures are six orders of magnitude too low (mPa vs kPa), and (3) the onset of sound production is three orders of magnitude too slow (ms vs µs).

## Dyad ratios are shared within each clade

We have previously reported (*Tobias and Kelley, 1987*) that sound pulses produced ex vivo in male *X. laevis* larynges include species-specific frequencies also present in actual calls. To determine whether spectral features of calls reflect species-typical laryngeal features, we first recorded male advertisement calls from representative *Xenopus* species. In the three clades of this sub-genus - L (which includes *X. laevis*), M (which includes *X. borealis*) and A (which includes *X. andrei* and *X. amieti*) (*Evans et al., 2015*) - each repeated sound pulse in the male advertisement call has two simultaneously produced frequency bands: a higher dominant frequency (DF2) and a lower frequency (DF1) (*Tobias et al., 2011*), termed dyads (*Figure 3A,C,D*). All but one species produce advertisement calls made up of harmonic dyads in which DF1 and DF2 are related by a small-integer ratio. The exception is *X. allofraseri* in which pulses contain harmonic stacks. There is a broad range

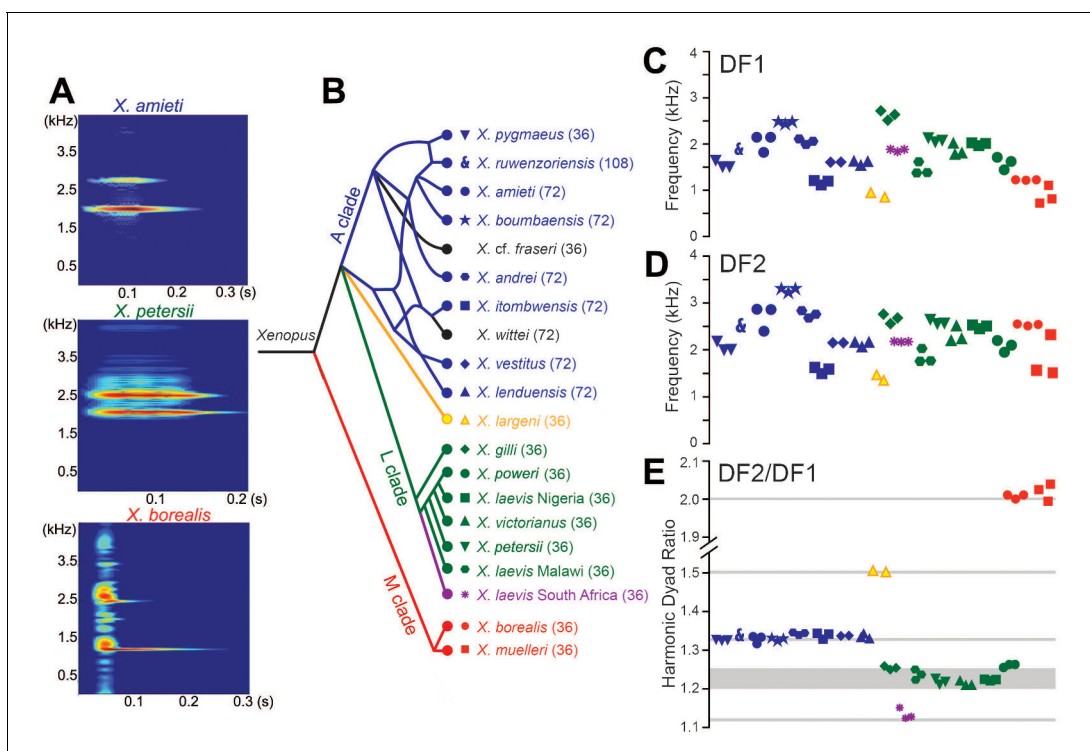

**Figure 3.** Harmonic dyad ratios are specific to, and highly conserved within, each *Xenopus* clade (blue: Clade A, green: Clade L; red: Clade M). (**A**) Each sound pulse in an advertisement call includes two dominant frequencies (DFs). Top: Spectrogram of multiple sound pulses in a *X. amieti* (A clade) advertisement call. Middle: multiple sound pulses in a *X. petersii* (L clade) advertisement call. Bottom: One sound pulse in a *X. borealis* (M clade) advertisement call. (**B**) Phylogenetic relationships of *Xenopus* species in this study. Clades and the DF2/DF1 ratios are: 1.33 (blue, A clade), 1.21–1.26 (green, L clade), and 2.0 (red, M clade). *X. allofraseri* (black) sound pulses are harmonic stacks. *X. laevis* South Africa (purple) and *X. wittei* (black) sound pulses are exceptions to their species group ratios. The ploidy level (number of chromosomes) is in parentheses; the DF2/DF1 (dyad) ratio for individual male calls for each species is indicated by a unique combination of symbol and color (**B–E**). The value of the lower dominant frequency (DF1; **C**) and the higher dominant frequency (DF2; **D**) respectively, in advertisement call sound pulses across *Xenopus*. (**E**) Harmonic dyad ratios fall into three major bands, one for each clade.

DOI: https://doi.org/10.7554/eLife.39946.008

for both the lower and the higher dominant frequency across *Xenopus* (*Figure 3C,D*). Either DF1 or DF2 can be shared by different species. However, in contrast to individual frequencies, the ratio of DF2/DF1 is specific to, and highly conserved within, each clade (*Figure 3E*). The mean ratio (*Table 1*) for A clade species is 1.33 (±0.004), for L clade species is 1.23 (±0.020), and for M clade species is 2.02 (±0.032). Exceptions include *X. wittei* in the A clade (1.19, not 1.34) and *X. laevis* South Africa in the L clade (1.14, not 1.24). Thus, species in different clades can share either DF1 or DF2 but not the distinctive ratio.

## Dyads are intrinsic to the larynx

To determine how dyads are produced, we first confirmed that our recordings of sound pulses were free of possible acoustic artifacts produced by interactions with the recording tank. Using a hydrophone and laser Doppler vibrometry, we obtained simultaneous recordings of sounds and body vibrations from a calling male under the same conditions in which we obtained the data in *Figure 3*. The same dyad was present in both sounds and vibrations from a vocalizing male (*Figure 4A*). Thus, the dyads reported here are produced by the frog. To determine whether dyads are produced solely by the larynx (without contributions from other organs), we recorded sound and vibrations produced by isolated larynges in response to nerve stimulation (*Figure 4B*). Isolated larynges produced the same dyads as the intact animal. Thus, the generation of dyads is intrinsic to the larynx and is not influenced by extra-laryngeal tissues.

To examine a potential role for the internal air spaces, we reduced the volume of the lumen in isolated *X. laevis* male larynges by inserting a large bead or replaced air with helium (as in a previous study; *Yager, 1992*). We also placed weights on dorsal surface of the larynx. None of these

**Table 1.** Spectral features of DF1 and DF2 across *Xenopus* species.
Average values (Standard Error). DF1: lower dominant frequency, DF2: higher dominant frequency. Additional broadband frequencies present at the attack of each sound pulse are not summarized here because they are not sustained. For both DF1 and DF2, the Q-value is the bandwidth at −6 dB divided by the peak frequency. Musical interval descriptors were assigned if the value of DF2/DF1 fell within 0.02 of a small integer ratio (e.g., 2:1 = Octave, 3:2 = Fifth, 4:3 = Perfect $4^{th}$, 5:4 = Major Third, 6:5 = Minor Third).

| Species | N | DF1 hz (SE) | DF1 Q-Value (SE) | DF2 hz (SE) | DF2 Q-Value (SE) | DF2/DF1 Ratio (SE) | Musical interval |
|---|---|---|---|---|---|---|---|
| *X. pygmaeus* | 3 | 1591 (40) | 16.6 (0.8) | 2124 (54) | 18.1 (2.6) | 1.34 (0.009) | Perfect 4th |
| *X. ruwenzoriensis* | 1 | 1937 | 21.6 | 2575 | 27.8 | 1.33 | Perfect 4th |
| *X. amieti* | 3 | 2043 (107) | 22.4 (1.3) | 2715 (152) | 27.2 (2.4) | 1.33 (0.005) | Perfect 4th |
| *X. boumbaensis* | 3 | 2475 (16) | 27.6 (0.2) | 3291 (29) | 33. 9 (0.4) | 1.33 (0.004) | Perfect 4th |
| *X. allofraseri* | 1 | N/A | N/A | N/A | N/A | N/A | Harmonic stack |
| *X. andrei* | 3 | 2062 (30) | 20.8 (1.5) | 2759 (51) | 26.7 (2.7) | 1.34 (0.006) | Perfect 4th |
| *X. itombwensis* | 3 | 1179 (32) | 12.4 (0.6) | 1573 (38) | 15.3 (2.1) | 1.33 (0.006) | Perfect 4th |
| *X. wittei* | 3 | 1319 (35) | 13.9 (0.7) | 1545 (38) | 15.0 (1.1) | 1.17 (0.003) | Non-consonant |
| *X. vestitus* | 2 | 1609 (2) | 17.7 (0.1) | 2150 (2) | 23.3 (0.3) | 1.34 (0.003) | Perfect 4th |
| *X. lenduensis* | 3 | 1595 (28) | 17.4 (0.2) | 2131 (33) | 22.8 (0.2) | 1.34 (0.004) | Perfect 4th |
| *X. largeni* | 2 | 929 (45) | 10.0 (0.2) | 1387 (59) | 14.7 (0.4) | 1.49 (0.010) | Perfect 5th |
| *X. gilli* | 3 | 2114 (45) | 22.7 (1.3) | 2641 (62) | 28.7 (1.3) | 1.25 (0.003) | Major Third |
| *X. poweri* | 3 | 1714 (28) | 19.1 (0.4) | 2159 (38) | 22.7 (0.1) | 1.26 (0.004) | Major Third |
| *X. laevis* Nigeria | 3 | 2029 (17) | 22.5 (0.2) | 2481 (22) | 26.6 (0.2) | 1.22 (0.004) | Minor Third |
| *X. victorianus* | 3 | 1891 (78) | 21.3 (0.9) | 2295 (102) | 22.4 (1.5) | 1.21 (0.005) | Minor Third |
| *X. petersii* | 3 | 2124 (24) | 23.6 (0.3) | 2576 (22) | 27.7 (0.3) | 1.21 (0.004) | Minor third |
| *X. laevis* Malawi | 3 | 1482 (80) | 15.8 (0.3) | 1831 (75) | 17.3 (0.7) | 1.24 (0.016) | Major Third |
| *X. laevis* South Africa | 3 | 1879 (21) | 16.0 (0.9) | 2154 (9) | 18.9 (2.2) | 1.15 (0.008) | Non-consonant |
| *X. borealis* | 3 | 1253 (5) | 13.9 (0.1) | 2504 (11) | 27.8 (0.4) | 2.00 (0.000) | Octave |
| *X. muelleri* | 2 | 1107 (29) | 12.2 (0.5) | 2262 (23) | 16.0 (6.1) | 2.04 (0.032) | Octave |

DOI: https://doi.org/10.7554/eLife.39946.009

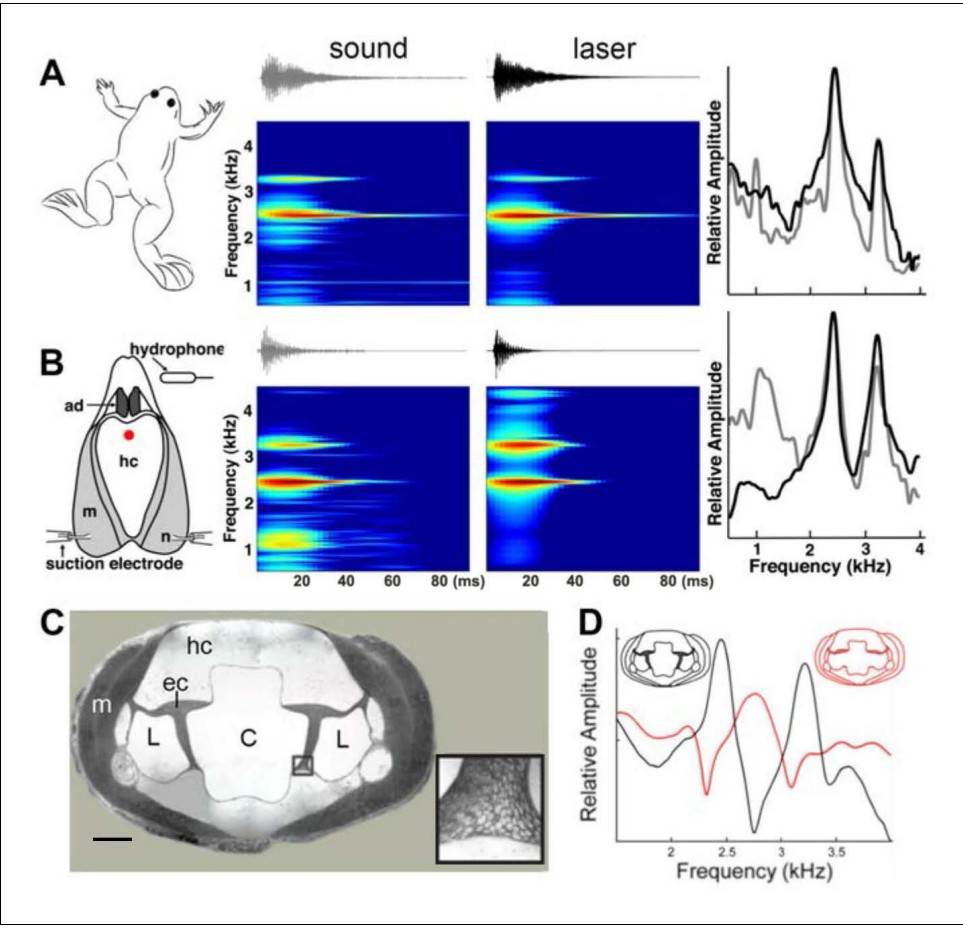

**Figure 4.** Harmonic dyad production is intrinsic to the isolated larynx and requires intact elastic cartilage septa. A singing male and an isolated larynx with intact elastic cartilage septa produce the same dyad in sound and laser recordings (A) Recordings from a *X. boumbaensis* male. Spectrograph (frequency vs time) of one pulse obtained from sound and laser vibrometry recordings. At right: Power spectra from the sound (grey) and laser (black) spectrographs. (B) Recordings from the isolated larynx of a *X. boumbaensis* male. Spectrograph (frequency vs time) of one pulse obtained from sound and laser vibrometry recordings, At right: Power spectra from the sound (grey) and laser (black) spectrographs. Note that the broad band frequency peak ~1 kHz in the sound recordings (gray) is not present in the laser recordings (black) and is thus an artifact of recording conditions (glass aquaria or Petri dish). Spectrographs are color graded to show increased intensity (blue to red). C - D Intact elastic cartilage is required for the production of frequency dyads. (C) Transverse section of an osmicated, epon-embedded male *X. laevis* larynx just anterior to the nerve (n) entry point in B; dorsal is up. The hyaline cartilage (hc) frame is flanked laterally by paired, bipennate laryngeal dilator muscles (m). Elastic cartilage sheets divide the lumen of the larynx into a central chamber (C) and two symmetrical lateral chambers (L). Elastic cartilage is recognizable by its unique, lace-like cellular morphology (inset). Scale bar 1 mm. (D) When elastic cartilage of the isolated larynx is disrupted bilaterally (red) via puncture (see *Figure 1—figure supplement 1*), the two DFs characteristic of the intact larynx (in black) are replaced by a single, intermediate DF in red.

DOI: https://doi.org/10.7554/eLife.39946.010

The following figure supplement is available for figure 4:

**Figure supplement 1.** Disruption of the elastic cartilage in an isolated larynx; the hole used to access arytenoid cartilages does not, by itself, disrupt dyads.

DOI: https://doi.org/10.7554/eLife.39946.011

manipulations affected DF1, DF2 or the DF2/DF1 ratio. These observations indicate that the dyads do not reflect the volume of the internal chambers or the mass of the cricoid cartilage. However, the ability of tissues containing elastic cartilage, such as the pinna of the ear, to deform and reform rapidly suggested that this tissue - also present in the larynx -might be essential to producing dyads.

The interior of the larynx (*Figure 4C*) includes a central, air-filled chamber (C) separated from smaller, lateral chambers (L) by elastic cartilage septa (*Figure 4C*, inset). We isolated the larynx and drilled a small hole into the middle of the dorsal hyaline cartilage through which a pin was used to puncture the elastic cartilage on both sides. After puncture, the elastic cartilage was no longer structurally intact (*Figure 4—figure supplement 1*). In the three species examined - *X. boumbaensis* (A clade; n = 6), *X. victorianus* (L clade, n = 2) and *X. laevis* South Africa (L clade- DF2/DF1 exception; n = 4) - the hole in the hyaline cartilage by itself (*Figure 4—figure supplement 1A*) did not affect the frequencies produced by motor nerve stimulation (*Figure 4—figure supplement 1B*)). However, puncturing the elastic cartilage (*Figure 4—figure supplement 1C*) either abolished the DF peaks by shifting the two narrowly tuned bands to a single, intermediate, broader band frequency (5/12 larynges; *Figure 4D*), abolished one peak and broadened the other (2/12), abolished both peaks (4/12) or shifted peaks to a different ratio (1/12). In *X. borealis,* 'opening' the cricoid box by removing a rectangular portion of the ventral laryngeal wall detunes the larynx (*Yager, 1992*). This detuning could also have been due to disruption of elastic cartilages. Because the elastic cartilage partitions the internal air chambers of the larynx, the puncture created a single air space.

## Discussion

Our data do not support the prevailing (*Yager, 1992*; *Irisarri et al., 2011*; *Ladich and Winkler, 2017*) cavitation hypothesis for sound production in *Xenopus*. Two alternative mechanisms that could account for the association between disc movement and sound production are: 1) acoustic excitation by a rapid pressure drop between the discs or 2) disc movement-associated mechanical excitation of the larynx. Sound pressure reduction between the discs may produce a propagating sound pressure wave, exciting air cavity resonances within the larynx. This mechanism should result in a bipole sound source with strong directional radiation pattern (*Larsen and Wahlberg, 2017*). Because of impedance mismatch between the air cavities and cartilages of the larynx, however, this mechanism would produce a poor, low intensity sound. In addition, replacing air with helium should alter the frequency distribution of cavity resonances (*Rand and Dudley, 1993*), an effect not observed in this study, nor previously (*Yager, 1992*).

Alternatively, separation of the arytenoid discs might mechanically excite vibration of laryngeal elements. This mechanism would result in a monopole sound source – the entire larynx - with a more omnidirectional radiation pattern, effectively coupled to the medium, producing a more intense sound. The sound pressure produced by such a vibrating monopole structure depends on its space and time averaged velocity (*Hambric and Fahnline, 2007*), which is consistent with our observations of a linear correlation between disc velocity and sound pressure, and the minimal velocity required for sound production. This mechanism is also consistent with previous experiments in which splitting 'the elastic sac surrounding the discs' (*Yager, 1992*) prevented sound pulse production. We thus favor the second explanation and propose that disc movements specifically excite vibration of the elastic tissues surrounding the discs (illustrated in *Figure 1—figure supplement 1*).

As key features of the *Xenopus* larynx - including lack of vocal folds and modification of the laryngeal box and cartilages - are shared across Pipid species (*Ridewood, 1897*; *Ridewood, 1900*), this proposed mechanism of underwater sound production may also be shared. In *Xenopus*, water is prevented from entering the larynx during underwater calling by inhibition of glottal motor neurons (*Zornik and Kelley, 2007*), thus ensuring the attainment of the disc peak velocity values identified here as required for sound production. The sound-protection afforded would not be required in another pipid, *Hymenochirus merlini*, that has reverted to calling in air (*Irisarri et al., 2011*), presumably through an open glottis. We predict that *H. merlini* calling is also powered by disc separation rather than air flow.

Our experimental results support the hypothesis that arytenoid disc movements subsequently excite two natural vibratory resonance frequencies of the larynx itself. These harmonic dyads require intact elastic cartilage septa that separate the central laryngeal lumen from the lateral laryngeal chambers. Both species-specific individual DFs and the clade-specific dyad ratio are thus intrinsic to the larynx rather than the result of laryngeal or respiratory muscle modulation by neural circuitry. Which, as yet unidentified, characteristics of laryngeal tissue geometry and properties result in species-specific DFs and their ratios remain to be determined, but are likely to reflect a common tuning mechanism in descendants of ancestral *Xenopus* species (*Sassoon et al., 1986*; *Baur et al., 2008*).

Three mechanisms have been identified for producing and shaping vertebrate laryngeal vocalizations: the myoelastic aerodynamic theory (MEAD), active muscle contraction (ACM) and intralaryngeal aerodynamic whistles. The MEAD mechanism (*Titze, 1980*) explains the physical basis for sounds produced by isolated larynges of mammals and terrestrial frogs (*Suthers et al., 2006*) as well as by syringes in birds (*Elemans et al., 2015*). The ACM mechanism requires motor neuron-driven contraction of intrinsic laryngeal muscles to produce, for example, purring in cats (*Remmers and Gautier, 1972*). Intralaryngeal aerodynamic whistles produce ultrasound in mice and probably all murine rodents (*Mahrt et al., 2016*; *Roberts, 1975*). All of these mechanisms require air flow to produce vocalizations and thus none of them explain how *Xenopus* call. The return to water in ancestral *Xenopus* was instead accompanied by a novel mechanism for laryngeal sound production: disc movement-induced excitation of laryngeally intrinsic resonance modes - dyads - shaped spectrally by material properties of elastic cartilage septa. Thus, the evolutionary change that allowed sound to be produced underwater without airflow in Pipids is not only responsible for production of sound pulses, but also their spectral features. Species-specific, rhythmic activity patterns of laryngeal motor neurons drive the precise temporal pattern of laryngeal muscle contractions responsible for the arytenoid disc separations that produce vocalizations (*Leininger and Kelley, 2013*; *Barkan et al., 2017*; *Barkan et al., 2018*).

In terrestrial frogs, as in other vocal vertebrates, acoustic features of male advertisement calls contain information on species identity (*Gerhardt and Huber, 2002*). This is also true for *Xenopus*; other call types - such as the male release call - vary little between species (*Tobias et al., 2014*). Information on species identity can serve to reduce interspecific mating and costs associated with hybrid offspring, including male sterility, that lead to restricted gene flow and speciation (*Lemmon and Lemmon, 2010*). Across *Xenopus*, temporal patterns of the advertisement call are homoplasious: the same pattern - for example a click-type call – recurs in genetically distant species (*Tobias et al., 2011*; *Roberts, 1975*). While temporal information can be ambiguous due to homoplasy, our data suggest that spectral information in advertisement calls – constrained by the morphology of the larynx – is more phylogenetically informative (*Gingras et al., 2013*).

However, *Xenopus* evolution has also been shaped by multiple rounds of inter-specific hybridization resulting in genomic introgression and the numerous highly polyploid species of the phylogeny, particularly A clade species (*Evans et al., 2015*). Rapid oviposition once eggs are ovulated places a premium on locating a male. When different species share the same pond, a female mating with a male from the same clade is more likely to produce viable and fertile offspring. The peripheral auditory system of females is tuned to their species' own dyad: DF1, DF2 and the DF2/DF1 ratio (*Hall et al., 2016*). Species-specific complementarity between vocal production and perception should reinforce the divergence of populations during speciation by limiting gene flow. The acoustic advantage to a gravid female of locating the most genetically-compatible calling male using the clade-specific common harmonic vocal signature thus may drive co-evolution of the vocal organ in the male and auditory perception in the female.

## Materials and methods

### Species

Subjects for this study were all sexually mature, male, clawed frogs from the sub-genus *Xenopus*. We either obtained frogs from commercial suppliers (Avifauna, Xenopus Express, Xenopus One, or Nasco) or from our *Xenopus* colony at Columbia University (species and populations: *X. pygmaeus*, *X. ruwensoriensis*, *X. amieti*, *X. boumbaensis*, *X. allofraseri*, *X. andrei*, *X. itombwensis*, *X. wittei*, *X. vestitus*, *X. lenduensis*, *X. largeni*, *X. gilli*, *X. poweri*, *X. laevis* Nigeria, *X. laevis* South Africa, *X. victorianus*, *X. petersii*, *X. laevis* Malawi, *X. borealis* and *X. muelleri*; see (*Tobias et al., 2011*) for details on geographic locales; species nomenclature as revised (*Evans et al., 2015*).) Frogs were housed in 2–5 L of water in polycarbonate tanks, under a 12–12 light-dark cycle, fed frog brittle (Nasco; Ft. Atkinson, WI, USA) and had their water changed twice per week. All animals were housed and handled in accordance with the guidelines established by the Danish Animal Experiments Inspectorate (Copenhagen, Denmark) and the Columbia University Animal Care and Use Committee.

## Measuring arytenoid disc acceleration and velocity; sound pulse production

We used high speed films of tissue movements in the isolated larynx preparation (*Tobias and Kelley, 1987*) of 5 adult male *X. laevis* to test the cavitation hypothesis. The animals were euthanized and their vocal organ and attached lungs were removed. The air-filled larynx was submerged in physiological saline in a 66 mm Petri dish. After isolation, larynges were pinned, dorsal side up, via extra-laryngeal cartilages, to a Sylgard coated recording dish submerged in oxygenated Ringers solution. The bilateral, freed motor nerves were drawn into suction electrodes for stimulation (WPI Linear stimulus isolator model A395R-C). All isolated larynges produced sound pulses.

Sound was recorded with a 1/2-inch pressure microphone-pre-amplifier assembly (model 46AD, G.R.A.S., Denmark) and amplified (model 12AQ, G.R.A.S., Denmark). The microphone and recording chain sensitivity was measured before each experiment (sound calibrator model 42AB, G.R.A.S., Denmark). The microphone was placed at 22–24 mm away from the mounted larynx. Because the signal-to-noise ratio of the hydrophone was lower than the microphone and the timing of sound events did not differ, we used the microphone signal for further analysis. Microphone, hydrophone and stimulation timing signals were low-pass filtered at 10 kHz, (custom-built filter, ThorLabs, Germany) and digitized at 30 kHz (USB 6259, 16 bit, National Instruments, Austin, Texas).

Larynges were imaged with a 12 bit high-speed camera (MotionPro-X4, 12 bit CMOS sensor, Integrated Design Tools, Inc.) mounted on a stereomicroscope (M165-FC, Leica Microsystems). The preparation was back-lighted to visualize the arytenoid discs by a plasma light source (HPLS200, Thorlabs, Germany) through liquid light guides and reflected of a 45° angled silver coated prism (MRA series, Thorlabs) to absorb heat. To track the position of landmarks, we placed 40–80 μm diameter carbon spheres on the surface of the larynx, muscles and arytenoid discs as illustrated in *Figure 1*. The position of spheres was tracked at subpixel precision by interpolating the 2D intensity cross-correlations of the same sphere in an initial frame to each movie image (*Figure 1B–D*). Velocity and acceleration of the spheres were calculated by differentiation of their position. All control and analysis software was written in MATLAB. In all five preparations, we filmed the larynx *in toto* following varying rates of nerve stimulation. In three preparations we obtained sufficiently high contrast images of the arytenoid discs at high imaging frame rates of 10,00–44,000 fps to allow automated position extraction during stimulation of the bilateral motor nerves at 40 Hz for 50 cycles. Data acquisition on the NI board and camera system was synchronized by a 1 ms TTL pulse. The camera was triggered at the positive rise of this 1 ms TTL pulse. The camera's specifications allow shutter speed as short as 1 μs. During earlier synchronization tests (*Elemans et al., 2015*), we determined that the trigger accuracy was below the duration of one frame (maximally 21 μs) and thus well below the relevant time scales investigated here.

Arytenoid gap width was defined as distance moved between the two markers from their resting position and perpendicular to the midline. Minimum and maximum acceleration of gap width were calculated per stimulus. Sound was bandpass filtered between 1–4 kHz (3th order butterworth filter implemented with zero phase-shift; filtfilt algorithm). The noise floor was defined as three times the standard deviation of a 67 ms background recording prior to each stimulation experiment. However, because sound energy did not fully dissipate in the experimental chamber between consecutive nerve stimulations, especially after 30–40 cycles, we used a threshold of 0.01 Pa to determine sound onset per stimulus. The first detectable sound pulse typically occurred after 2–3 stimulations. This is consistent with our earlier work showing that male laryngeal neuromuscular synapses are 'weak,' i. e., require facilitation to release sufficient neurotransmitter to generate a muscle action potential and contraction (*Ruel et al., 1997*). We used linear regressions - including only measurements above sound threshold – to calculate the minimal disc velocity and acceleration associated with sound generation.

## Recordings of vocal behavior

For in vivo recordings, frogs were placed in a 75 L aquarium. To promote vocal behavior, males were injected with human chorionic gonadotropin (hCG; Sigma: 50–200 IU depending on body size) one day prior to and on the day of recording. Males were then paired with a conspecific, sexually unreceptive female in a glass aquarium (60 × 15 × 30.5 cm, L × W × H; water depth = 23 cm; 20∘ C). A hydrophone (High Tech, Gulfport, MI, USA; output sensitivity −164.5 dB at 1 V/μPa, frequency

sensitivity 0.015–10 kHz; or Cornell Bioacoustics, output sensitivity −163 dB at 1 V/μPa) was used to record calls to a Marantz digital recorder CD or flash card (CDR300, Marantz, Mahwah, NJ, USA; 44.1 kHz sampling rate) or on a computer (Macintosh) via a Lexicon A/D converter. To measure values for DF1 and DF2, three non-consecutive advertisement calls (the smallest vocal unit as described in (Tobias et al., 2011)) were analyzed from 3 males of each species. Dominant frequencies were calculated from fast Fourier transforms (FFT) with maximum Q values (peak frequency/maximum frequency 6 dB below peak frequency - minimum frequency 6 dB below peak frequency; Table 1). The initial attack segment of each sound pulse was not included in the analysis because it is more broadband than the sustained portion of the call. The values shown in Table 1 are the mean of individual means for all calls by species recorded.

## Sound and laser recordings in vivo and ex vivo

Advertisement calls of single males were recorded in aquaria with a hydrophone (H2a, Aquarian Audio Products; Anacortes, WA, USA) and vibration velocities were recorded simultaneously with a portable laser (PDV 100 laser, Polytec Inc.; Irvine, CA, USA) directed at the ventral surface of the singing frog. We recorded from one each of *X. laevis.* South Africa, *X. borealis*, *X. muelleri*, *X. new tetraploid*, and *X. boumbaensis.* We then isolated the larynx as described above. To access the elastic cartilages (*Figure 4—figure supplement 1C*), we drilled a small hole in the dorsal surface of the larynx (*Figure 4—figure supplement 1A*) and then sealed it with a small piece of Parafilm. To ensure that the hole had no effect, sound and laser recordings were obtained before and after this procedure (*Figure 4—figure supplement 1B*). The Parafilm was then removed and a 30 g needle used to puncture the elastic cartilage on both sides, after which the Parafilm was replaced. At the end of the experiment, the larynx was split saggitally ('butterflied') and the disruption of elastic cartilage was confirmed by visual inspection (*Figure 4—figure supplement 1C*, illustrating intact vs punctured elastic cartilage). Sound and laser recordings were digitized (PreSonus Audio box; Baton Rouge, LA, USA) and stored on a Macintosh computer.

## Acknowledgements

This work was supported by a National Institutes of Health grant (R01 NS23684) and research funds associated with the Weintraub Chair (DBK), the Danish Research Council (FNU) and Carlsberg Foundation awards to CPHE, post-doctoral fellowships from the NIH (F32 GM103266) and the Revson Foundation to ICH and Amgen and SURF awards to UKB. We thank Carolyn Diaz for laryngeal histology, Avelyne Villain for statistical analyses, Irene Ballagh and Charlotte Barkan for *Figure 1A*, Erik Zornik and Charlotte Barkan for reviewing the manuscript, and Sheila Patek, Jakob Christensen-Dalsgaard and Ron Hoy for advice.

## Additional information

### Funding

| Funder | Grant reference number | Author |
| --- | --- | --- |
| Amgen Foundation | | Ursula Kwong-Brown |
| National Institutes of Health | F32 GM103266 | Ian C Hall |
| Charles H. Revson Foundation | | Ian C Hall |
| Carlsbergfondet | | Coen PH Elemans |
| Det Frie Forskningsråd, Natur og Univers | Sapere Aude 2 | Coen PH Elemans |
| National Institutes of Health | NS23684 | Darcy B Kelley |

The funders had no role in study design, data collection and interpretation, or the decision to submit the work for publication.

## Author contributions

Ursula Kwong-Brown, Conceptualization, Resources, Data curation, Formal analysis, Supervision, Funding acquisition, Investigation, Visualization, Methodology, Writing—original draft, Project administration, Writing—review and editing; Martha L Tobias, Conceptualization, Data curation, Formal analysis, Supervision, Investigation, Visualization, Methodology, Project administration, Writing—review and editing; Damian O Elias, Conceptualization, Formal analysis, Supervision, Investigation, Visualization, Methodology, Writing—original draft, Project administration, Writing—review and editing; Ian C Hall, Conceptualization, Investigation, Visualization, Writing—review and editing; Coen PH Elemans, Formal analysis, Investigation, Visualization, Writing—review and editing; Darcy B Kelley, Conceptualization, Resources, Data curation, Software, Formal analysis, Supervision, Funding acquisition, Investigation, Visualization, Methodology, Writing—original draft, Project administration, Writing—review and editing

## Author ORCIDs

Ursula Kwong-Brown (iD) http://orcid.org/0000-0002-8099-2649
Coen PH Elemans (iD) http://orcid.org/0000-0001-6306-5715
Darcy B Kelley (iD) http://orcid.org/0000-0003-4736-4939

## Ethics

Animal experimentation: This study was performed in strict accordance with the recommendations in the Guide for the Care and Use of Laboratory Animals of the National Institutes of Health. All of the animals were handled, sound recordings acquired and tissues collected according to approved institutional animal care and use committee (IACUC) protocols of Columbia University (AC-AAAE1004; New York, NY, USA). High speed filming of isolated larynges were carried out in accordance with the Danish Animal Experiments Inspectorate ( license #2013-15-2934-00991;Copenhagen, Denmark)

## Decision letter and Author response

Decision letter https://doi.org/10.7554/eLife.39946.016
Author response https://doi.org/10.7554/eLife.39946.017

# Additional files

## Supplementary files

• Transparent reporting form
DOI: https://doi.org/10.7554/eLife.39946.012

## Data availability

Imaging data have been made available using an appropriate repository (Dryad). Examples of male advertisement calls from each species were deposited in AmphibiaWeb by MLT (https://amphibiaweb.org/lists/Pipidae.shtml).

The following dataset was generated:

| Author(s) | Year | Dataset title | Dataset URL | Database and Identifier |
|---|---|---|---|---|
| Ursula Kwong-Brown, Martha L Tobias, Damian O Elias, Ian C Hall, Coen PH Elemans, Darcy B Kelley | 2018 | Data from: The return to water in ancestral Xenopus was accompanied by a novel mechanism for producing and shaping vocal signals | https://doi.org/10.5061/dryad.220602k | Dryad Digital Repository, 10.5061/dryad.220602k |

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
