## [Decision Letter]

[**Editorial note:** This article has been through an editorial process in which the authors decide how to respond to the issues raised during peer review. The Reviewing Editor's assessment is that all the issues have been addressed.]

Thank you for submitting your article "The return to water in ancestral *Xenopus* was accompanied by a novel mechanism for producing and shaping vocal signals" for consideration by *eLife*. Your article has been reviewed by three peer reviewers, including Min Zhu as the Reviewing Editor and Reviewer #1, and the evaluation has been overseen by Andrew King as the Senior Editor. The following individuals involved in review of your submission have also agreed to reveal their identity: Franz Goller (Reviewer #2); Fernando Montealegre-Zapata (Reviewer #3).

The Reviewing Editor has highlighted the concerns that require revision and/or responses, and we have included the separate reviews below for your consideration. If you have any questions, please do not hesitate to contact us.

Summary:

This is a high-quality study that explores the sound generating mechanism in the unusual larynx of *Xenopus*, a clade of frogs mostly calling under water. The results refute the current hypothesis for *Xenopus* vocalization, cavitation, and suggest sound production by mechanical excitation of laryngeal resonance modes following rapid separation of laryngeal arytenoid discs. The study also shows that the resulting frequency modes (dyads) are intrinsic to the larynx, and provides new insights in how essential acoustic information for social communication in frogs can be linked to laryngeal structure. As such, this study will be very interesting both to the scientific community and to the public.

Major concerns:

1) To help the readability of your hypothesis, the text figures should be improved in labellings. There also exist the figure citation errors.

Figure legends, "Figure 1—figure supplement 2" should be "supplement 1". In the main-text, the citation of "Figure 1—figure supplement 2" is now in front of that of "Figure 1—figure supplement 1".

Figure 1—figure supplement 2: The labeling and legend explanation need clarification (ad vs ac), what is it? Figure 2—figure supplement 1 seems to show the opposite of what was stated for water injection in regard to velocity and acceleration.

Figure 1: panel A does not correspond to the title of this panel the figure caption: *Xenopus* call while submerged. Panel B, the isolated larynx, will be benefit from having body coordinates labels (anterior, posterior, etc). This is for the non-experts. I also find it very distracting the use of acronyms in the labels. Why don't use fine arrows and the full name of the structure in white fonts? Or make the anatomy panels bigger?

Figure 2. Panel A. Difficult so distinguish between the blue and green of the stimuli. Correct caption: (color coded as inl A)

Figure 3. Panel A needs time units. Figure will also benefits from labelling clades L, M, X.

Figure 4. A) there is no red spot on the frog shown in panel A. Time axis of the spectrograms needs units. D) "see Supplementary Figure SI1" to be clarified.

2) Please detail some methods in sound and laser analysis. The readers will benefit to know how the high-speed video was synchronized with data acquisition.

3) We are wondering whether these data allow the authors to discern between the respective roles of excursion amplitude of the discs and peak velocity/acceleration in determining the threshold for sound detection. Is it really peak velocity and acceleration that determine whether or not sound is detectable or does the actual amplitude of excursion of the discs also play a role? A larger excursion could generate larger pressure peaks and only then is sound detectable.

4) All the genus and species names in the main-text and references should be in italics. Some references such as Larsen and Larsen (2017) should be edited in the right format.

Separate reviews (please respond to each point):

*Reviewer #1:*

The transitions at the edge of water in vertebrates are among the critical topics of evolutionary biology. Most tetrapods use air to produce sound in specilized vocal organs such as the larynx of frogs and mammals and the syrinx of birds. Some groups of tetrapods, such as pipid frogs, cetaceans and extinct marine reptiles, however returned to fully aquatic lifestyles. How these land vertebrates adapted their originally air-driven sound production system for social communication under water remains ambiguous. This study explores how air-driven sound production system changed when the land ancestor of aquatic frog *Xenopus* returned to water. The results refute the current hypothesis for *Xenopus* vocalization, cavitation, and suggest sound production by mechanical excitation of laryngeal resonance modes following rapid separation of laryngeal arytenoid discs. The study also shows that the resulting frequency modes (dyads) are intrinsic to the larynx, and provides new insights in how essential acoustic information for social communication in frogs can be linked to laryngeal structure. As such, this study will be very interesting both to the scientific community and to the public.

Minor Comments:

In the section "references", all the genus and species names should be in italics. Some references such as Larsen and Larsen (2017) should be edited in the right format.

Figure legends, "Figure 1—figure supplement 2" should be "Supplement 1". In the main-text, the citation of "Figure 1—figure supplement 2" is now in front of that of "Figure 1—figure supplement 1".

Figure 4: D) "see Supplementary Figure SI1" to be clarified.

*Reviewer #2:*

General comment:

This is an elegant study focused on the sound generating mechanism in the unusual larynx of *Xenopus*, a clade of frogs mostly calling under water. The data are convincing and the phylogenetic distribution of the ratio of dyad frequencies suggests a morphological basis for species identity. Unfortunately, the precise origin of this dyad signature remains to be discovered. I have a few suggestions, comments and questions, which are detailed below.

Abstract:

I would argue that cavitation is hardly a theory, rather a proposed mechanism and as such a hypothesis (as correctly stated in the Introduction).

*Xenopus* should be italicized.

Introduction:

The point that we do not know in sufficient detail how specific acoustic signatures are generated is well taken. The transition to the return to water however may have provided a simple case in which this can be studied. The special mechanism however does not inform us how it is done in a vocal fold vibration mechanism.

Results:

I am wondering whether these data allow the authors to discern between the respective roles of excursion amplitude of the discs and peak velocity/acceleration in determining the threshold for sound detection. Is it really peak velocity and acceleration that determine whether or not sound is detectable or does the actual amplitude of excursion of the discs also play a role? A larger excursion could generate larger pressure peaks and only then is sound detectable.

Figure 2—figure supplement 1 seems to show the opposite of what was stated for water injection in regard to velocity and acceleration.

Discussion:

The production mechanism for ultrasound in rodents is currently not clear. In fact, the mechanism cited here (Mahrt et al., 2016) has been credibly challenged (Riede et al., 2017). Although not central to the current analysis, it would be appropriate to cite all papers here.

The last paragraph does not seem very relevant to the story presented here.

Figure 1—figure supplement 2:

The labeling and legend explanation need clarification (ad vs ac), what is t?,

*Reviewer #3:*

This is an outstanding interesting piece of work, and I really enjoyed reading it. The authors took an interdisciplinary approach to answer a challenging question about sound production in aquatic frogs. The research is innovative as the team used a combination of state-of-the-art technologies (HS video, Laser Doppler vibrometry, condenser mics and hydrophones recordings, electrophysiology, comparative analysis, etc) to answer the questions. Sound production in aquatic frogs is not well understood and has previously been explained by a mechanism known as 'implosion of air bubbles or cavitation'. The authors challenged this explanation and more convincingly demonstrated that bubbles or high-velocity-flow are never formed during the process, and instead their data supports another mechanism: sound production by mechanical excitation of laryngeal resonance modes. The authors took this analysis further and investigated this across several species of the genus *Xenopus*. Results imply that this mechanism is shared across the species studied, therefore its relevance in various fields, including evolutionary biology.

I have some comments, which I consider minor, and which I hope the team will take into consideration to improve the clarity of the paper, etc

Introduction:

Paragraph 1: very clear, and targets the point.

Paragraph 2: Some bits of the description of the anatomy here are far more complex than the respective Figure For example, the anatomy of the vocal organ and nasal and buccal cavities is not clear in panel A of Figure 1. Fonts are too small and only brain and larynx are labelled; please label the other structures as well (although many aspects are clarified in Figure 1 supplement). For Other suggestion see comments on Figure 1, below. I had some problems imagining how this disk moves, and this is due to the indistinguishable anatomy of the AC in the inset of Figure 1A.

Paragraph 3: What is a high velocity separation of the AD? You mean high velocity motion produces air bubbles? The film of fluid between the AD should be label, I supposed in Figure 1—figure supplement.2?

Results:

Check figure suggestions.

Dyads are intrinsic to the larynx: The laser and mic approach is an ingenious way to investigate how dyads are produced.

Materials and methods:

Animal handling and experimentation followed ethics protocols.

Disk acceleration and velocity: Experimental procedures were necessary; sample low, but the complexity of the experiments and the fact that subjects are vertebrates justifies it.

Subsection “Measuring arytenoid disc acceleration and velocity; sound pulse production”; second paragraph. This part will benefit from more details on the use of each microphone and hydrophone, so that a sequence of the events could be replicated. SNR, please explain acronym.

Third paragraph in the same subsection: In 3 preparations (in three preparations).

Also in the third paragraph: “Arytenoid gap width was defined as distance moved between the two markers from their resting position and perpendicular to the midline.” Can this be pointed in a figure? Also, a large number of people understand SPL better than pressure in Pa, perhaps more appropriate to use SLP here for a broader audience?

Subsection “Recordings of vocal behavior”, High-speed video (supplementary material). A reader will benefit to know how the high-speed video was synchronized with data acquisition.

A few spacing typos needed in this paragraph (oron, Tomeasure, thesmallestvocalunitsdescribedin), check the rest.

Subsection “Sound and laser recordings in vivo and ex vivo”: Sound and laser analysis. The combination of laser and sound recordings is a good approach to answer your questions convincingly. I have however a couple minor questions here: 1) you report pressure units (Pa), but it is not clear how the microphone was calibrated. You mention the GRAS calibrator, but somewhere an interface to insert a correction value to obtain an amplitude of 1.0024 Pa at 1kHz should be available (I supposed in a Matlab program). For all of us in the field this a routing procedure, but students and people outside the field would probably benefit from such details. 2) The laser Doppler vibrometer you used could measure velocity, displacement or acceleration, please describe which of these were utilized.

Discussion:

Third paragraph:....across Pipid species (semicolon).

Later: "thus ensuring the attainment of the disc acceleration values required for sound production identified here." I find this sentence challenging to digest, please improve.

Figures:

Figure 1: panel A does not correspond to the title of this panel the figure caption: *Xenopus* call while submerged. Panel B, the isolated larynx, will be benefit from having body coordinates labels (anterior, posterior, etc). This is for the non-experts. I also find it very distracting the use of acronyms in the labels. Why don't use fine arrows and the full name of the structure in white fonts? Or make the anatomy panels bigger?]

Figure 1—figure supplement 2: label 't' is not described in the caption.

Figure 2. Panel A. Difficult so distinguish between the blue and green of the stimuli. Correct caption: (color coded as inl A)

Figure 3. Panel A needs time units. Figure will also benefits from labelling clades L, M, X.

Figure 4. A) there is no red spot on the frog shown in panel A. Time axis of the spectrograms needs units.

---

## [Author Response]

Major concerns:1) To help the readability of your hypothesis, the text figures should be improved in labellings. There also exist the figure citation errors.Figure legends, "Figure 1—figure supplement 2" should be "supplement 1". In the main-text, the citation of "Figure 1—figure supplement 2" is now in front of that of "Figure 1—figure supplement 1".

We have changed all the references to the supplementary figures to match their corresponding figure legends exactly. Figure 1—figure supplement 1 illustrates the elastic cartilage surrounding the arytenoid discs.

Figure 1—figure supplement 2: The labeling and legend explanation need clarification (ad vs ac), what is t?

The paper includes (in order of appearance): Figure 1 and Figure 1—figure supplement 1, Video 1, Figure 2 and Figure 2—figure supplement 1, Figure 3, Figure 3—figure supplement 1 and Figure 4 and Figure 4—figure supplement 1.

We have expanded the legend for Figure 1—figure supplement 1 as follows: "Transverse section (5 μm) through a decalcified. osmicated, epon-embedded larynx of a male *X. laevis* at the anterior-posterior level of the arytenoid discs (ad, Figure 1B inset); dorsal is up. As in *X. borealis,*10 the anterior arytenoid cartilages (ac) and arytenoid discs (ad) are suspended in elastic tissue including elastic cartilage (ec) identifiable by its characteristic "Swiss-cheese" appearance (inset at left). Seams of ec also insert bilaterally onto the cricoid box (cricoid), composed of hyaline cartilage, and form the septa between the lateral and medial chambers more posteriorly (illustrated in Figure 4C). Laryngeal muscles (m) insert via the tendon (t) onto the arytenoid discs just posterior to the level of this section (see Figure 1B inset)."

Figure 2—figure supplement 1 seems to show the opposite of what was stated for water injection in regard to velocity and acceleration.

Yes, this was incorrect; see response below.

Figure 1: panel A does not correspond to the title of this panel the figure caption: Xenopus call while submerged.

We have expanded the figure legend to address these concerns as follows:

Figure 1 Arytenoid disc kinematics associated with underwater sound production in the ex vivolarynx of *Xenopus laevis*. A) *Xenopus* call while submerged. A ventral view of a reproductively active, *X. laevis* male (nuptial pads in grey on the inner surface of the forearms), underwater (blue waves); larynx in red and more dorsal brain in blue. This view of the larynx is schematic (i.e. the dorsal rather than the ventral side is illustrated) in order to correspond to the actual isolated larynx in (B). On the left, an oscillogram (sound intensity vs time) of a single, biphasic call that includes a fast and slow trill. Each vertical line indicates a sound pulse; ~60 pulses/s for fast trill and ~30 pulses/s for slow trill.

Panel B, the isolated larynx, will be benefit from having body coordinates labels (anterior, posterior, etc). This is for the non-experts.

Figure 1A (above) now provides anatomical landmarks (e.g. anterior: head, posterior: hind limbs) for the reader who should now be able to match the larynx *in situ* in the frog in A, to the picture in panel B. This view of the larynx is schematic (i.e. the dorsal rather than the ventral side is illustrated) in order to correspond to the actual isolated larynx in (B).

I also find it very distracting the use of acronyms in the labels. Why don't use fine arrows and the full name of the structure in white fonts? Or make the anatomy panels bigger?

We have made the whole figure bigger and the names of all the laryngeal components in Figure 1 panel B are now spelled out, in full, in the left-hand portion of that panel. One fine black line points to the arytenoid cartilage and another to some of the carbon microspheres. A fine white line points out one nerve in a suction electrode. We however continue to use initials in the inset for panel B, illustrating the arytenoids discs (ad) within the arytenoid cartilages (ac) and the tendon (t), to avoid obscuring these components. This information has been added to the Figure legend as follows:

“B) Dorsal aspect of an isolated X. laevis larynx, a cricoid box of hyaline cartilage flanked by muscles. Each effective contraction/relaxation of these paired laryngeal muscles produces a single sound pulse. In the preparation illustrated, sound pulses are evoked by electrical stimulation of both laryngeal nerves via suction electrodes. Inset: Each muscle contraction produces a transient increase in tension on the arytenoid discs (ad) located within the arytenoid cartilages (ac) via the tendons (t). Globule cells (gc) secrete a mucopolysaccharide onto the medial surfaces of the arytenoid discs. 10 Carbon microspheres (e.g. M1 and M2) placed on the surface of the larynx track muscle and cartilage positions.”

Figure 2. Panel A. Difficult so distinguish between the blue and green of the stimuli. Correct caption: (color coded as inl A)

We have changed the color coding of responses to successive stimuli from the original green to blue progression, to an aqua to blue progression. Color saturation now increases as sound pulse intensity increases, which should be more easily distinguished even by readers with blue-yellow color blindness. The caption has been corrected and now reads:"(color coded as in A)"

Figure 3. Panel A needs time units. Figure will also benefits from labelling clades L, M, X.

We have added time units to panel A and labelled the clades in panel B.

Figure 4. A) there is no red spot on the frog shown in panel A. Time axis of the spectrograms needs units. D) "see Supplementary Figure SI1" to be clarified.

We have removed "Red spot indicates the location of the laser recording." from the figure legend and added time units to the spectrograms. We have corrected the reference to the Supplementary Figure.

2) Please detail some methods in sound and laser analysis. The readers will benefit to know how the high-speed video was synchronized with data acquisition.

We have added this information to the Materials and methods section: "Data acquisition on the NI board and camera system was synchronized by a 1 ms TTL pulse. The camera was triggered at the positive rise of this 1 ms TTL pulse. The camera’s specifications allow shutter speed as short as 1 μs. During earlier synchronization tests we determined that the trigger accuracy was below the duration of one frame and maximally 21 μs2 and thus well below the relevant time scales investigated here."

3) We are wondering whether these data allow the authors to discern between the respective roles of excursion amplitude of the discs and peak velocity/acceleration in determining the threshold for sound detection. Is it really peak velocity and acceleration that determine whether or not sound is detectable or does the actual amplitude of excursion of the discs also play a role? A larger excursion could generate larger pressure peaks and only then is sound detectable.

Detection of sound pulses depends on the signal-to-noise ratio of the sound pressure recording; low amplitude pulses can be masked by noise. Because the sound energy remains in the experimental chamber for several milliseconds after a sound pulse, accurate onset of sound detection as nerve stimulation proceeds causes the noise level to increase and this affects sound onset detection accuracy. We therefore set the threshold for sound detection rather conservatively as described in the Materials and methods:

"The noise floor was defined as three times the standard deviation of a 67ms background recording prior to each stimulation experiment. However, because sound energy did not fully dissipate in the experimental chamber between consecutive nerve stimulations, especially after 30-40 cycles, we used a threshold of 0.01 Pa to determine sound onset per stimulus."

Before any sound pulses have been generated, noise is about 5 mPa ptp; the first detectable sound pulse typically occurred after 2-3 stimulations. This result is consistent with our earlier work showing that male laryngeal neuromuscular synapses are "weak" i.e. require facilitation to release sufficient neurotransmitter to generate a muscle action potential and contraction. We have added this information to the Materials and methods (and provide a correct version of the appropriate reference.15)

Our hypothesis is that the mucopolysaccharide liquid coating the medial surfaces of the discs forms an adhesive liquid bridge. The discs separate when force exerted on the discs by the muscles, via the tendon, overcomes the adhesive liquid bridge force. In general, the respective roles of position, velocity and acceleration of disc kinematics can be hard to dissect, because all correlate positively with sound amplitude (e.g. Figure 2 and Figure 2—figure supplement 1). However introducing water prevented the liquid bridge formation and shed some light on this question.

In our revision we added gap width, next to disc peak velocity and peak acceleration, to Figure 2—figure supplement 1. These data clearly show that after adding water in the gap, the discs still move and form a gap up to 80 μm – exceeding the 50 μm amplitude that accompanies sound pulses produced before water was introduced. The lower maximal disc excursion is most likely due to the fact that less energy is injected in the spring system by the muscles because the discs are not kept together by the liquid bridge force. Of disc gap width, peak velocity and peak acceleration values associated with sound pulse production, only disc peak velocity does not reach threshold when the liquid bridge holding the discs together is disrupted (Figure 2—figure supplement 1). This observation supports the hypothesis that a threshold disc peak velocity is required for sound production.

The manuscript has been revised to include the position data in Figure 2—figure supplement 1 and we added the following text:: "Of disc gap width, peak velocity and peak acceleration values associated with sound pulse production, only disc peak velocity does not reach threshold when the liquid bridge holding the discs together is disrupted (Figure 2—figure supplement 1). This observation supports the hypothesis that a threshold disc peak velocity is required for sound production."

4) All the genus and species names in the main-text and references should be in italics.

*Xenopus* in the text (opening of the last paragraph) is now italicized as are all genus and species names in the References.

Some references such as Larsen and Larsen (2017) should be edited in the right format.

We have corrected the Larsen and Larsen reference and reviewed the formatting of the References.

Separate reviews (please respond to each point):

Reviewer #1:

[…] The study also shows that the resulting frequency modes (dyads) are intrinsic to the larynx, and provides new insights in how essential acoustic information for social communication in frogs can be linked to laryngeal structure. As such, this study will be very interesting both to the scientific community and to the public.Minor Comments:In the section "references", all the genus and species names should be in italics. Some references such as Larsen and Larsen (2017) should be edited in the right format.

See response above

Figure legends, "Figure 1—figure supplement 2" should be "supplement 1". In the main-text, the citation of "Figure 1—figure supplement 2" is now in front of that of "Figure 1—figure supplement 1".

See response above

Figure 4: D) "see Supplementary Figure SI1" to be clarified.

See response above.

Reviewer #2:

General comment:This is an elegant study focused on the sound generating mechanism in the unusual larynx of Xenopus, a clade of frogs mostly calling under water. The data are convincing and the phylogenetic distribution of the ratio of dyad frequencies suggests a morphological basis for species identity. Unfortunately, the precise origin of this dyad signature remains to be discovered. I have a few suggestions, comments and questions, which are detailed below.Abstract:I would argue that cavitation is hardly a theory, rather a proposed mechanism and as such a hypothesis (as correctly stated in the Introduction).

"theory" has been changed to "mechanism" in the Introduction.

Xenopus should be italicized.

See response above.

Introduction:The point that we do not know in sufficient detail how specific acoustic signatures are generated is well taken. The transition to the return to water however may have provided a simple case in which this can be studied. The special mechanism however does not inform us how it is done in a vocal fold vibration mechanism.

A number of studies in terrestrial frogs have examined the role of air-driven vibration of the vocal folds (cords) in generating sounds (references 12, 17, 24 and 31). About 120 years ago, Ridewood reported that the Pipids lack vocal folds. and speculated about how sounds might be produced, underwater, without them. The sounds that make up species-specific vocal communication in anurans are important in communicating essential social information including species, sex, age, reproductive state, rivalry. In our studies, we asked how this information could be still be produced in *Xenopus* without air flow and vocal folds. Our evidence supports the idea that the Pipid return to water from land was accompanied by a new way of making sound that preserves the kinds of acoustic information that was ancestrally essential. We note that another Pipid, *Hymenochirus boettergi*, has returned to land from water and that movement of air from the lungs into the mouth cavity accompanies call production (reference 11) and predict this new way of producing sounds persists, an idea which can be tested directly.

Results:I am wondering whether these data allow the authors to discern between the respective roles of excursion amplitude of the discs and peak velocity/acceleration in determining the threshold for sound detection. Is it really peak velocity and acceleration that determine whether or not sound is detectable or does the actual amplitude of excursion of the discs also play a role? A larger excursion could generate larger pressure peaks and only then is sound detectable.

See response above.

Figure 2—figure supplement 1 seems to show the opposite of what was stated for water injection in regard to velocity and acceleration.

See response above.

Discussion:The production mechanism for ultrasound in rodents is currently not clear. In fact, the mechanism cited here (Mahrt et al., 2016) has been credibly challenged (Riede et al., 2017). Although not central to the current analysis, it would be appropriate to cite all papers here.

The data presented in the Mahrt et al. paper questioned a prevailing hypothesis that ultrasonic vocalizations (USVs) are generated by hole-tone whistles or superficial vocal fold movements. The authors proposed that rodent USVs are instead produced by feedback between downstream convecting coherent flow structures from the glottis and upstream-propagating acoustic waves, perhaps also due to reverberating acoustic conditions. However, they conclude that: “What exact mechanism constitutes the feedback remains unknown; it could be either upstream propagating acoustic disturbances through the core of the jet, or outside of the jet, or some edge effect.”

The wall or edge tone is hard to separate and still rather intensively debated in mechanical engineering literature. The equations predicting stable modes are exactly the same for both core, wall or edge feedback mechanisms. The paper by Riede et al., 2017 shows that several rodent species have an intralaryngeal ventral pouch that affects USVs. While their data inform the question of what geometric structure could provide feedback, results do not challenge the main conclusions of Mahrt et al. Though very interesting, this discussion is not relevant for the main conclusions of our paper.

The last paragraph does not seem very relevant to the story presented here.

Species-specific courtship songs generally serve as pre-zygotic barriers that prevent mating between males and females of different species. The consequences of choosing a different species to mate with is producing hybrid offspring that are less successful than pure species offspring, either in viability or reproduction. All *Xenopus* species, however, arose from ancient intra-specific hybridizations. L clade species produce fertile hybrids of both sexes that can mate with each other but hybrids between L and M clade species do not produce inter-fertile hybrids. Recognizing a potential reproductive partner from the same clade (*i.e.* genetically similar) is thus advantageous and we propose that this recognition drives the remarkable conservation of the dyad (DF2/DF1 ratio) within each clade. The dyad ratio provides a feature of voice that *Xenopus* can use locate a genetically favorable partner. We have revised the paragraph for clarity as follows:

“However, *Xenopus* evolution has also been shaped by multiple rounds of inter-specific hybridization resulting in genomic introgression and the numerous highly polyploid species of the phylogeny, particularly A clade species.4 Rapid oviposition once eggs are ovulated places a premium on locating a male. When different species share the same pond, a female mating with a male from the same clade is more likely to produce viable and fertile offspring. The peripheral auditory system of females is tuned to their species' own dyad: DF1, DF2 and the DF2/DF1 ratio.35 Species-specific complementarity between vocal production and perception should reinforce the divergence of populations during speciation by limiting gene flow. The acoustic advantage to a gravid female of locating the most genetically compatible calling male using the clade-specific common harmonic vocal signature thus may drive co-evolution of the vocal organ in the male and auditory perception in the female.”

Figure 1—figure supplement 2:The labeling and legend explanation need clarification (ad vs ac), what is t?,

See response above.

Reviewer #3:

[…] I have some comments, which I consider minor, and which I hope the team will take into consideration to improve the clarity of the paper, etcIntroduction:Paragraph 1: very clear, and targets the point.Paragraph 2: Some bits of the description of the anatomy here are far more complex than the respective Figure For example, the anatomy of the vocal organ and nasal and buccal cavities is not clear in panel A of Figure 1. Fonts are too small and only brain and larynx are labelled; please label the other structures as well (although many aspects are clarified in Figure 1 supplement). For Other suggestion see comments on Figure 1, below. I had some problems imagining how this disk moves, and this is due to the indistinguishable anatomy of the AC in the inset of Figure 1A.

See response above.

Paragraph 3: What is a high velocity separation of the AD? You mean high velocity motion produces air bubbles? The film of fluid between the AD should be label, I supposed in Figure 1—figure supplement 2?

The velocity of disc separation should be clear in the revised version of Sl Figure 2. We did not observe the air bubbles that would be predicted if sound production is due to cavitation. The film is due to secretion of the goblet cells into the space above the arytenoid discs (gc in Figure 1 inset).

Results:Check figure suggestions.

We have followed suggestions for revised figures.

Dyads are intrinsic to the larynx: The laser and mic approach is an ingenious way to investigate how dyads are produced.Materials and methods:Animal handling and experimentation followed ethics protocols.Disk acceleration and velocity: Experimental procedures were necessary; sample low, but the complexity of the experiments and the fact that subjects are vertebrates justifies it.Subsection “Measuring arytenoid disc acceleration and velocity; sound pulse production”; second paragraph. This part will benefit from more details on the use of each microphone and hydrophone, so that a sequence of the events could be replicated.

This information is provided.

SNR, please explain acronym.

The acronym has been deleted

Third paragraph in the same subsection: In 3 preparations (in three preparations).

Five and three are now spelled out.

Also in the third paragraph: “Arytenoid gap width was defined as distance moved between the two markers from their resting position and perpendicular to the midline.” Can this be pointed in a figure?

Figure 2—figure supplement 1 now provides information on position. The maximum gap is illustrated (aqua line) in Figure 1C.

Also, a large number of people understand SPL better than pressure in Pa, perhaps more appropriate to use SLP here for a broader audience?

In order to report in SI units we use Pa (Pascals).

Subsection “Recordings of vocal behavior”, High-speed video (supplementary material). A reader will benefit to know how the high-speed video was synchronized with data acquisition.

See response above.

A few spacing typos needed in this paragraph (oron, Tomeasure, thesmallestvocalunitsdescribedin), check the rest.

Fine in the actual document, must have happened in the conversion to PDF.

*Subsection “Sound and laser recordings* in vivo *and* ex vivo*”: Sound and laser analysis. The combination of laser and sound recordings is a good approach to answer your questions convincingly. I have however a couple minor questions here: 1) you report pressure units (Pa), but it is not clear how the microphone was calibrated. You mention the GRAS calibrator, but somewhere an interface to insert a correction value to obtain an amplitude of 1.0024 Pa at 1kHz should be available (I supposed in a Matlab program). For all of us in the field this a routing procedure, but students and people outside the field would probably benefit from such details.*

The GRAS calibrator was used for the microphone recordings in air; we provide information on calibration. Laser measurements were combined with simultaneous hydrophone recordings (underwater in singing males).

2) The laser Doppler vibrometer you used could measure velocity, displacement or acceleration, please describe which of these were utilized.

It measured velocity; added to the text.

Discussion:Third paragraph:.….across Pipid species (semicolon).

Why a semicolon? The sentence reads: "As key features of the *Xenopus* larynx -including lack of vocal folds and modification of the laryngeal box and cartilages – are shared across Pipid species, 8,19 this proposed mechanism of underwater sound production may also be shared."

Later: "thus ensuring the attainment of the disc acceleration values required for sound production identified here." I find this sentence challenging to digest, please improve.

Now revised.

Figures:Figure 1: panel A does not correspond to the title of this panel the figure caption: Xenopus call while submerged. Panel B, the isolated larynx, will be benefit from having body coordinates labels (anterior, posterior, etc). This is for the non-experts. I also find it very distracting the use of acronyms in the labels. Why don't use fine arrows and the full name of the structure in white fonts? Or make the anatomy panels bigger?]

See response above.

Figure 1—figure supplement 2: label 't' is not described in the caption.

Now corrected.

Figure 2. Panel A. Difficult so distinguish between the blue and green of the stimuli. Correct caption: (color coded as inl A)Figure 3. Panel A needs time units. Figure will also benefits from labelling clades L, M, X.Figure 4. A) there is no red spot on the frog shown in panel A. Time axis of the spectrograms needs units.

See responses above.